# SRDGAN: LEARNING THE NOISE PRIOR FOR SUPER RESOLUTION WITH DUAL GENERATIVE ADVERSARIAL NETWORKS

## ABSTRACT

Single Image Super Resolution (SISR) is the task of producing a high resolution (HR) image from a given low-resolution (LR) image. It is a well researched problem with extensive commercial applications like digital camera, video compression, medical imaging, etc. Most recent super resolution works focus on the feature learning architecture, like Chao Dong (2016); Dong et al. (2016); Wang et al. (2018b); Ledig et al. (2017). However, these works suffer from the following challenges: (1) The low-resolution (LR) training images are artificially synthesized using HR images with bicubic downsampling, which have much more information than real demosaic-upscaled images. The mismatch between training and realistic mobile data heavily blocks the effect on practical SR problem. (2) These methods cannot effectively handle the blind distortions during super resolution in practical applications. In this work, an end-to-end novel framework, including high-to-low network and low-to-high network, is proposed to solve the above problems with dual Generative Adversarial Networks (GAN). First, the above mismatch problems are well explored with the high-to-low network, where clear high-resolution image and the corresponding realistic low-resolution image pairs can be generated. With high-to-low network, a large-scale **General Mobile Super Resolution Dataset, GMSR,** is proposed, which can be utilized for training or as a benchmark for super resolution methods. Second, an effective low-to-high network (super resolution network) is proposed in the framework. Benefiting from the GMSR dataset and novel training strategies, the proposed super resolution model can effectively handle detail recovery and denoising at the same time.

## 1 INTRODUCTION

This paper targets on single image super resolution (SISR). Various works have been done to recover the details from low-resolution image and generate realistic high-resolution image. Recently, learning based methods have shown great potential in solving image processing problems. Zhang et al. (2016; 2015); Guan et al. (2017); Guan & Cham (2017). Especially, development of deep learning provides various learning-based solutions for super resolution Chao Dong (2016); Dong et al. (2016); Ledig et al. (2017); Wang et al. (2018b). Among the learning-based methods, generative adversarial network (GAN), Goodfellow et al. (2014), achieved great progress. Ledig et al. (2017); Wang et al. (2018b) proved that GAN-based methods can generate plausible-looking natural details which are consistent with the human visual system (HVS).

However, mismatches exist between the data sets utilized by most existing learning-based methods and the real images captured by a mobile phone. First, in most super resolution methods, low-resolution(LR) images are simply downsampled from the high-resolution image with a fixed known method, for example bicubic interpolation. As a result, the informative level of the low-resolution image is actually 'richer' than the image size. However, In most practical case, Bayer pattern decides that one pixel only collect intensity for one color channel, and the left channels are usually upscaled by demosaicing. Thus, the information of mobile real-taken images is under-informative than its actual image size. Therefore, the informative level mismatch exists between the training LR data and the mobile captured data. Second, the LR image generated by downsampling is clear and with no artifacts, while the LR image captured by mobile suffers from many distortions, such as

noise and blur, as shown in Fig. 1 (a). The distortion mismatch also exists. These mismatches between the downsampled LR training data and real LR data (mobile-captured data) heavily restrict the performance of these learning-based algorithms. As shown in Fig. 1 (b), the HR image generated by the state-of-the-art method ESRGAN Wang et al. (2018b) cannot handle real mobile images well.

The other way is using optical methods to generate HR and LR image pairs, like Zhang et al. (2019). However, it suffers many problems, such as mismatches of angle, lighting, depth-of-field and object position, which is hard to avoid. Besides, the whole process is time-consuming, thus it is hard to generate a very large-scale dataset. Therefore, an end-to-end framework is proposed in this paper, which consists of two parts. First, a high-to-low network $\Phi_{H2L}$ is trained to generate realistic HR/LR image pair for training super resolution models, named **GMSR**. This dataset can promote fair comparison and support further research on the SR problems in general. Second, a low-to-high network $\Phi_{L2H}$ is trained to produce the super resolution image. Novel training methods were utilized to reproduce better results. As shown in Fig. 1 (c), the HR image generated by our method can handle the noise better than ESRGAN as illustrated by the blue patch. Moreover, better details can be reproduced as shown in the red patch. To sum up, the proposed SR method can effectively deal with the real distortions and restore fine details.

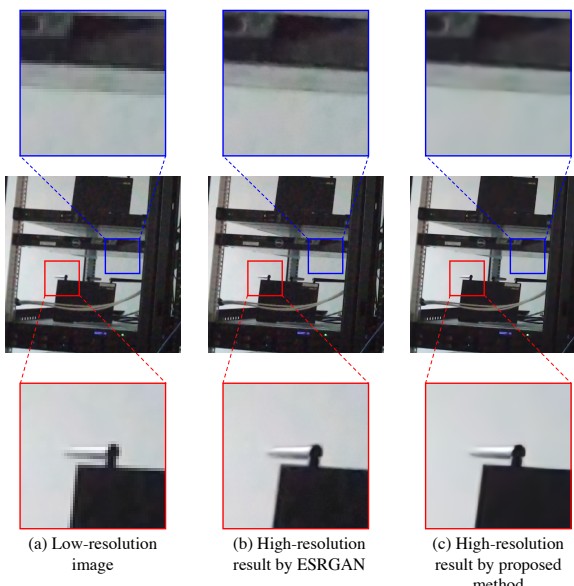

(a) Low-resolution image    (b) High-resolution result by ESRGAN    (c) High-resolution result by proposed method

Figure 1: (a) Example of an mobile low-resolution image, which is utilized as the input of super resolution (SR) methods. (b) Examples of super-resolution result via state-of-the-art method ESRGAN Wang et al. (2018b). (c) Examples of SR result via the proposed method. (Best viewed in color.)

The contributions of our work can be summarized into three categories. First, a general SR problem is explored which straightly face the mismatch between real mobile images and commonly utilized training data. A H2L network is proposed to solve the problems effectively. Second, an end-to-end framework is proposed, which includes two parts, high-to-low network (H2L) and super-resolution network (L2H). During training, dual Generative Adversarial Network (GAN) is utilized to optimize the parameters.Third, a large-scale General Mobile Super Resolution dataset, **GMSR**, is generated by high-to-low network. This dataset can support further research on the super resolution domain, and can be utilized either for training or as a fair benchmark.

## 2 RELATED WORK

Single image super resolution (SISR) is an important topic and has been developed for a long time. Early methods Li & Orchard (2001); Allebach & Wong (1996); Zhang & Wu (2006); Duchon (1979) that are based on the interpolation theory can be very fast, however usually yield over-smooth results. Methods rely on neighbor embedding and sparse representation Chang et al. (2004); Kim & Kwon (2010); Timofte et al. (2013; 2014); Yang et al. (2010); Zeyde et al. (2010a) targeted at learning the mapping between LR and HR. Some example-based approaches used image self-similarity property to reduce the amount of training data needed Glasner et al. (2009); Freedman & Fattal (2011); Gao et al. (2012), and increased the size of the limited internal dictionary Huang et al. (2015).

With the development of deep learning technology, methods based on convolution neural network have shown great potential in solving super resolution problems. Chao Dong (2016) first proposed an end-to-end convolutional neural network to learn the mapping between HR and LR. Kim et al. (2016a) improved the reconstruction accuracy by using very deep convolutional network and residual learning, and they further utilized recursive structure and skip-connection to improve perfor-

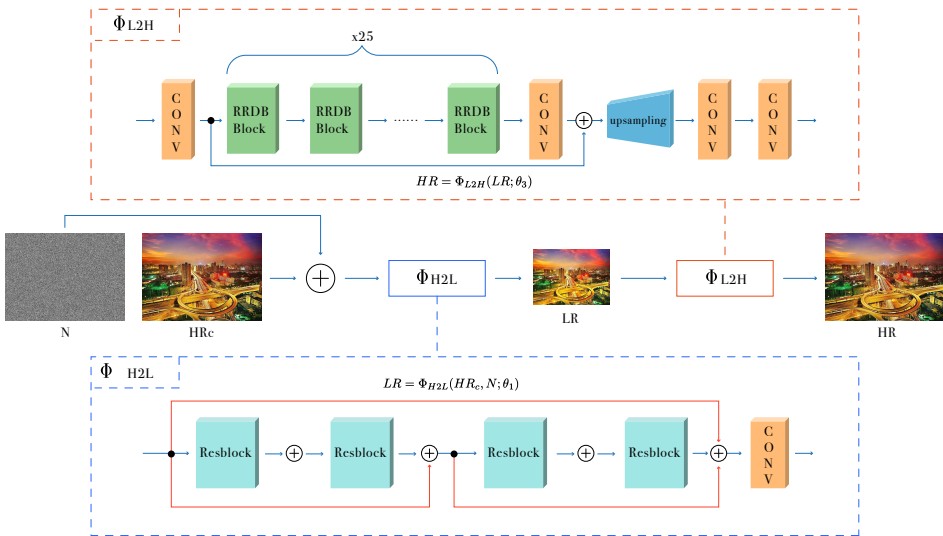

Figure 2: Pipeline of the proposed SRDGAN framework, including $\Phi_{H2L}$ and $\Phi_{L2H}$. The high-to-low network $\Phi_{H2L}$ is proposed to generate realistic HR/LR image pairs. The low-to-high network $\Phi_{L2H}$ is proposed to produce the super resolution image.

mance without introducing new parameters Kim et al. (2016b). To increase inference efficiency, methods with low-resolution images as input Lai et al. (2017); Dong et al. (2016); Shi et al. (2016); Lim et al. (2017) were proposed. In Lai et al. (2017), the authors proposed the Laplacian Pyramid network structure to reduce the computational complexity and improve the performance. Dong et al. (2016) and Shi et al. (2016) introduced different deconvolution methods to upscale the LR to HR.

Most recently, GAN based super resolution methods have been proposed and achieved great progress. Ledig et al. (2017) first applied GAN to super resolution task and got highly photo-realistic results. Wang et al. (2018b) improved the performance by modifying the network structure and loss based on SRGAN. Network conditioning was used in Wang et al. (2018a) to combine the category prior with the generative network to get more realistic textures. Yuan et al. (2018) developed unsupervised learning with GAN, where unpaired HR-LR data were utilized. Until now, most super resolution methods use the bicubic downsampled LR image as training data. However, since bicubic downsampling is quiet different from the degradation in real-world, it makes the trained model not effective to the real-world cases. Inspired by ESRGAN Wang et al. (2018b), We followed its main structure and proposed a deeper network named the low-to-high(L2H) network to solve the more complicated real-world cases, and build a large-scale General Mobile Super Resolution Dataset **GMSR**, which is potentially applicable to other image reconstruction methods. Bulat et al. (2018) adopted a degradation network to generate LR image first, then use it to train the model and get good result. However, they focus on face, which has much less complexity and diversity than general images. Hence, we extended their work to the general image super resolution. We proposed a high-to-low(H2L) network to learn the degradation from HR image to LR image, so that the 'realistic' LR image can be generated once we have the HR image. Zhao et al. (2018) proposed a degradation network to generate 'realistic' LR image and use it to train the degradation and SR construction network. However, they only use one discriminative network to train the degradation network. Different from them, two discriminative networks were utilized to train H2L and L2H network respectively in this work. Moreover, the two networks were jointly optimized as well.

## 3 PROPOSED FRAMEWORK

The proposed framework aims at solving real-world super resolution problems. In the framework, a high-to-low network is first proposed to generate realistic image pairs, which is trained with GAN using real-word images captured by mobiles. (Section 3.2). Secondly, a super resolution method

Table 1: Notations used in the paper.

| | |
|---|---|
| $\Phi_{H2L}$ | High-to-low Network |
| $I_{pair}=\{HR_c, HR_n\}$ | paired clear image $HR_c$ and noisy image $HR_n$ |
| $LR_n$ | Low-resolution image downsampled from $HR_n$ |
| $I_n$ | Mobile captured noisy images. No corresponding clear images |
| $I_{nc}$ | Cropped LR image from $I_n$ |
| **GMSR** | Proposed general mobile super resolution dataset |
| $I_{sim}$ | Image pairs in dataset **GMSR** |
| $HR_c$ | Clear high-resolution images as input of $\Phi_{H2L}$ |
| N | Random noise element-wised added to $HR_c$ |
| $\Phi_{L2H}$ | Low-to-high Network, i.e. super-resolution network |

with a novel structure and training strategy is explored (Section 3.3). Besides, new datasets are captured or generated, which are also introduced in this section.

## 3.1 DATASETS

Three novel datasets were captured and generated to train the proposed framework, including $I_{pari}$, $I_n$ and **GMSR**. $I_{pair}$ and $I_n$ were captured to train the $\Phi_{H2L}$ network. Dataset **GMSR** is simulated by $\Phi_{H2L}$ and utilized to train the super resolution model $\Phi_{L2H}$. Details are given as follows.

1. In order to learn the noise distribution of real world images, we applied smart phone to capture the dataset, named $I_{pair} = \{HR_c, HR_n\}$. Blackberry Key2 is selected to do this task due to its representative camera module. The resolution of images in $I_{pair}$ are $4032 \times 3024$. It contains 447 pairs of images, which are randomly divided into training, validation and test set with number 400, 27 and 20 respectively. Each pair contains a noisy image $HR_n$ and a corresponding clear image $HR_c$. To generate the clear image $HR_c$, a burst of 20 noisy images were captured with the mobile phone fixed on a tripod.

   A multi-frame denoising method was used to generate a clear high-resolution image $HR_c$ by adaptively fusing the 20 noisy images. Since every noisy images can be base image for fusion, 20 pairs of images can be generated for each scene. During training, $LR_n$ is down-sampled from $HR_n$, and constitutes an image pair with $HR_c$.

2. Dataset $I_n$ contain 206 captured noisy image by Blackberry key2 with resolution $4032 \times 3024$. Contents of the images are different from the ones in $I_{pair}$. We set $I_n$ as the reference image for the discriminative network to help the high-to-low network to learn the actual noise distribution.

3. General Mobile Super resolution **GMSR** is generated by $\Phi_{H2L}$ and contains 1153 image pairs. $I_{sim}$ represent image pairs in dataset **GMSR**. The HR images in **GMSR** are collected from the Internet.

## 3.2 HIGH-TO-LOW NETWORK

The difficulty of artificially generating HR-LR image pairs is mainly two-fold. One is to model distortions in LR images, the other is to generate image pairs with pixel-wise correspondence. To solve the problems, a high-to-low network $\Phi_{H2L}$ is proposed. Notations used in this paper are summarized in Table 1. LR image is generated based on a clear $HR_c$ as follows:

$$LR = \Phi_{H2L}(HR_c, N; \theta_1) \tag{1}$$

where $HR_c$ is a clear high-resolution image, the generated low-resolution image $LR$ is supposed to have similar noise distribution with real-world image. Gaussian random noise $N$ is added to simulate the randomness of the distortions in LR. The mean and standard deviation of the Gaussian noise is 0 and 0.05, respectively. The two values can be adjusted according to the distortion level of different mobiles. The noise transition between Gaussian distribution and that in real-world images are accomplished by $\Phi_{H2L}$ and learned through GAN.

**(a) Train** $\Phi_{H2L}$        **(b) Train** $\Phi_{L2H}$

Figure 3: Training procedure of the proposed framework with dual GAN. (a) introduces the training process of $\Phi_{H2L}$, (b) is the training procedure of $\Phi_{L2H}$.

### 3.2.1 MODEL STRUCTURE

$\Phi_{H2L}$ is able to model the image distortion of real mobile images caused in the image capturing and processing pipeline. The structure of $\Phi_{H2L}$ network is shown in Fig. 2, which consists of 4 Resblocks and 1 convolutional layer. Each resblock consists of two convolution layers and two activation functions. In order to further build the global connection Ahn et al. (2018), multiple shortcut connections is added as indicated by the red lines. The final convolution layer aims at decreasing the resolution by setting the downscale factor as the stride.

### 3.2.2 TRAINING PROCEDURE OF $\Phi_{H2L}$

The training strategy of $\Phi_{H2L}$ is summarized in Fig. 3. During the training process, dataset $I_{pair} = \{HR_n, HR_c\}$ and $I_n$ are used. In order to get the ground-truth of $LR$, noisy image $HR_n$ is downsampled to generate $LR_n$, which is used to calculate the pixel and feature losses. However, the noise distribution of $LR_n$ is not consistent with $HR_n$ after the down-sampling. Therefore, Generative Adversarial Network (GAN) is used to further learn the noise distribution of mobile images. The loss function and training strategy are introduced as follows.

**Loss function**: Pixel loss, feature loss (i.e. perceptual loss) and GAN loss are used to train $\Phi_{H2L}$.

$$loss_{H2L} = \alpha_1(loss^p_{H2L}) + \alpha_2(loss^f_{H2L}) + \alpha_3(loss^{GAN}_{H2L}) \qquad (2)$$

where $\alpha_1$, $\alpha_2$ and $\alpha_3$ are weighting parameters.

Pixel loss $loss^p_{H2L}$ aims at pixel-wisely comparing the image difference between $LR_n$ and the network output $LR$, as

$$loss^p_{H2L} = \frac{1}{n^p_{LR}} \sum_{k=1}^{n^p_{LR}} (\||LR(k) - LR_n(k)\||^2_2) \qquad (3)$$

where $n^p_{LR}$ corresponds to the number of pixels in $LR$.

Feature loss aims at comparing the difference between high-level features of two images, which has been proved to be effective in many previous works Ledig et al. (2017); Wang et al. (2018b); He et al. (2016); Johnson et al. (2016). Inspired by these works, VGG Simonyan & Zisserman (2014) is utilized as feature extraction network $\varphi$:

$$loss^f_{H2L} = \frac{1}{n^f_{LR}} \sum_{k=1}^{n^f_{LR}} (\||\varphi(LR)(k) - \varphi(LR_n)(k)\||_1) \qquad (4)$$

Where $\varphi(LR)$ is the extracted high-level features of input image LR by $\varphi$. $k$ represents the index in the extracted features. GAN is then used to learn the distribution by discriminating unpaired images, $I_{nc}$ and $LR_n$, where $I_{nc}$ is the low-resolution image randomly cropped from $I_n$.

$$loss^{GAN}_{H2L} = \frac{1}{n^{GAN}_{LR}} \sum_{n=1}^{n^{GAN}_{LR}} -logD_{\theta_2}(\Phi_{H2L}(HR_n, N)) \qquad (5)$$

where $n_{LR}^{GAN}$ is the number of batch size, $D_{\theta_2}$ is the discriminative model with parameters $\theta_2$. The goal of $D_{\theta_2}$ is to well distinguish the generated LR image and real image patch $I_{nc}$. $D_{\theta_2}$ is trained iteratively with the generative model $\Phi_{H2L}$ as shown in Equ. 6.

$$min_{\theta_1}max_{\theta_2}\mathbb{E}[logD_{\theta_2}(I_{nc})]+ \\ \mathbb{E}[log(1-D_{\theta_2}(\Phi_{H2L}(HR_N,N;\theta_1)))] \tag{6}$$

In this work, nearest neighbour is explored as the down-sampling method to better simulate the real-world ill-informative situation of mobile images.

### 3.3 LOW-TO-HIGH NETWORK

Network $\Phi_{L2H}$ is proposed to implement super resolution based on the low-resolution image $LR$ with parameter $\theta_3$ as in Equation 7.

$$HR = \Phi_{L2H}(LR;\theta_3) \tag{7}$$

#### 3.3.1 NETWORK STRUCTURE

The structure of $\Phi_{L2H}$ is shown in Fig. 2. RRDB block from Wang et al. (2018b) is used as the basic network unit. a deeper structure with 25 RRDB blocks is proposed to model the more complex task. The basic unit is not limited to RRDB block, other basic block like residual block and dense block can also work in the proposed network structure.

#### 3.3.2 TRAINING PROCEDURE OF $\Phi_{L2H}$

The training strategy of $\Phi_{L2H}$ is summarized in Fig. 3. The loss function of $\Phi_{L2H}$ is similar to that of $\Phi_{H2L}$, which is described in Section 3.2.2.

$$loss_{L2H} = \alpha_4(loss_{L2H}^p) + \alpha_5(loss_{L2H}^f) + \alpha_6(loss_{L2H}^{GAN}) \tag{8}$$

where $\alpha_4$, $\alpha_5$ and $\alpha_6$ are weighting parameters. The main difference of the training strategy between $\Phi_{H2L}$ and $\Phi_{L2H}$ is training data. For $\Phi_{L2H}$, two categories of training samples are utilized. The most important one is $I_{sim}$ from dataset GMSR that simulates real-world distortions. Furthermore, the popular dataset DIV2K Timofte et al. (2017) is utilized to further increase the data diversity.

Finally, global fine-tuning is performed on the whole network structure including $\Phi_{H2L}$, $\Phi_{L2H}$ and the discriminative networks, to jointly optimize the performance.

## 4 EXPERIMENTS

Eexperiments were conducted to evaluate the effectiveness of the whole proposed framework and its constituents including the H2L network and the training strategy. The parameters for training the proposed framework are described as follows. Batch-size is 16 with training patch of size $192 \times 192$.

Table 2: Quantitative evaluation of SR algorithms (PSNR/SSIM)

| Method | Set5 | Set14 | Set $I_{pair}$(test) |
|--------|------|-------|------------------|
| Bicubic | 28.42/0.8104 | 26.00/0.7027 | 35.85/0.9674 |
| SRCNN | 30.07/0.8627 | **27.50**/0.7513 | 39.09/0.9716 |
| SRGAN | 29.40/0.8472 | 26.02/0.7397 | 38.43/0.9596 |
| ESRGAN | **30.55**/0.8677 | 26.39/0.7246 | 38.55/0.9572 |
| Ours | 30.37/**0.8766** | 27.35/**0.7767** | **39.93/0.9782** |

Following ESRGAN Wang et al. (2018b), the learning rate is initialized with $1 \times 10^{-4}$, and halved at [50k, 100k, 200k, 300k] iterations. AdamKingma & Ba (2014) optimizer is utilized with $\beta_1 = 0.9$, $\beta_2 = 0.999$ without weight decay.

### 4.1 SUPER RESOLUTION RESULTS

Quantitative and Subjective experiments were conducted to evaluate the performance of the proposed method. Table 2 shows the quantitative evaluation results of the popular and state-of-the-art GAN-based super resolution methods with an upscale factor of 4.

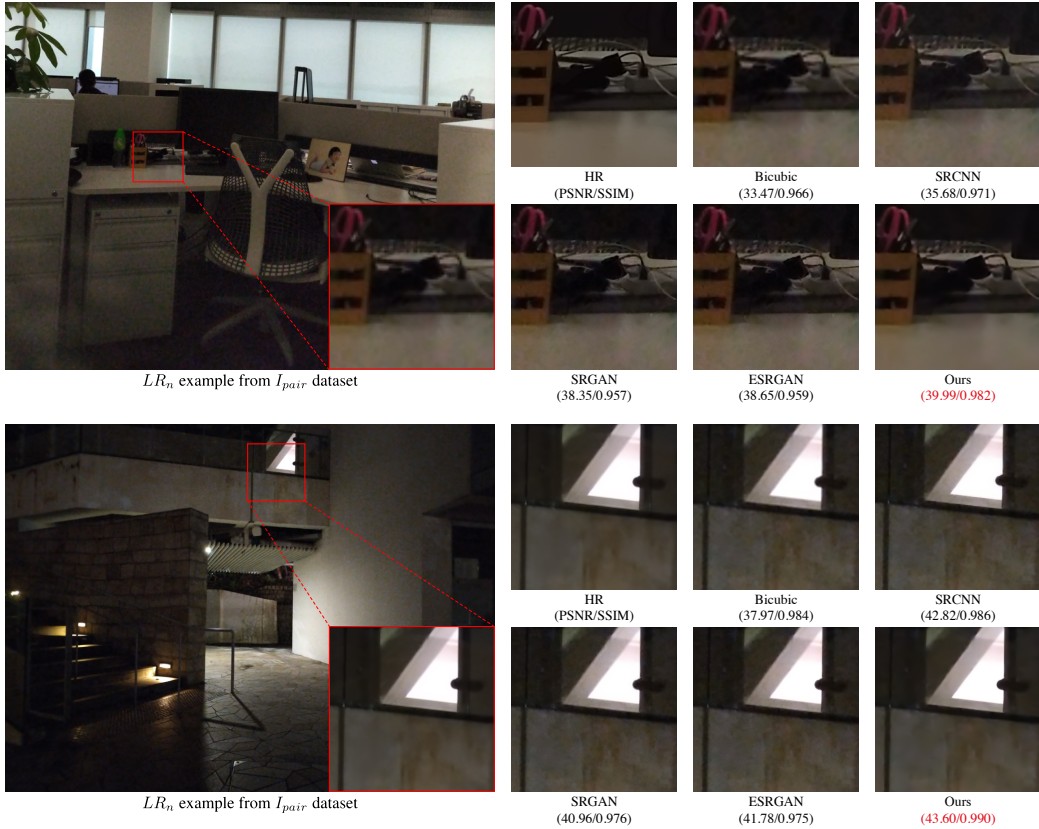

Figure 4: Subjective comparison between the proposed method and some state-of-the-art methods. Left is the mobile low-resolution image example and a detail patch. Right is the super-resolution result of different methods and high-resolution ground-truth (HR). (Best viewed in color)

Both general datasets (set5 Bevilacqua et al. (2012), set14 Zeyde et al. (2010b)) and the mobile-specific dataset ($I_{pair}$) are utilized. PSNR and SSIM Wang et al. (2004) are adopted for measuring the effectiveness of the proposed method, where higher value in-

Table 3: Subjective evaluation on real mobile images

| Mobile | Iphone7p | Xiaomi | IphoneXs | All |
|--------|----------|--------|----------|-----|
| Bicubic | 1.78 | 1.35 | 1.69 | 1.61 |
| ESRGAN | 1.83 | 2.31 | 1.98 | 2.04 |
| Ours | **2.39** | **2.34** | **2.32** | **2.35** |

dicates better performance. From Table 2, it is shown that the proposed method outperforms the other state-of-the-art methods in terms of SSIM in all circumstances, demonstrating its effectiveness on reconstructing structure. Especially, the improvements achieved on Set $I_{pair}$ are significant, showing its good performance on mobile images. The best performing results are shown in bold. Subjective comparison examples were presented as shown in Fig. 4. Real-world mobile images are utilized. PSNR and SSIM results are presented at the bottom for reference, which are evaluated on Y channel. In many situations, mobile images suffer from noise and detail missing. Most methods cannot handle these cases well, including bicubic, SRCNN Chao Dong (2016), SRGAN Ledig et al. (2017), ESRGAN Wang et al. (2018b). The noise is still very obvious, and even increased in SR results. Instead, our method can deliver good results by removing all the noise and well-reserve the edges and details.

Moreover, in order to verify the generalization ability of the proposed method, subjective evaluation experiments were conducted based on real phone-captured images. 24 images captured by three different mobile phones (Iphone7p, Xiaomi Mix2s and Iphone Xs Max) were used as inputs. We compared the performance of three SR methods: Bicubic, ESRGAN and the proposed SRDGAN. Totally 21 subjects without expert knowledge participated the experiment, who evaluated each up-sampled image subjectively using a score ranging from one to three. Higher score represents better

visual quality. The experimental results are shown in Table 3, which confirms that in the real-world case, the proposed method is applicable, and it largely outperforms Bicubic and the state-of-the-art method ESRGAN.

## 4.2 RESULTS OF $\Phi_{H2L}$

With $\Phi_{H2L}$, we aim to simulate LR images that is a better ensemble of real mobile images and to generate a large-scale dataset. To be fair, a training dataset that pre-processed with artificial degradation methods is gen-

Table 4:
Quantitative evaluation of $SR_{bl}$ and ours algorithms(PSNR/SSIM)

| Method | Set5 | Set14 | Set $I_{pair}$(test) |
|---|---|---|---|
| $SR_{bl}$ | 29.72/0.8404 | 27.05/0.7357 | 39.04/0.9736 |
| Ours | **30.37/0.8766** | **27.35/0.7767** | **39.93/0.9782** |

erated and serves as baseline. Specifically, high-quality HR images of GMSR dataset and DIV2K are downsampled with bicubic interpolation, Gaussian blurred with random intensity to simulate out-of-focus, converted into Bayer data, added with Gaussian noise, and finally debayer-filtered to generate the LR images. We used this baseline dataset to train the proposed SR model and named the new super resolution model $SR_{bl}$. Quantitative results are shown in Table 4. Our model trained on GMSR outperforms $SR_{bl}$ in all cases, demonstrating the effectiveness of the proposed H2L network and the GMSR dataset.

## 4.3 EXPLORING TRAINING STRATEGIES FOR $\Phi_{L2H}$

Experiments were conducted to evaluate the effectiveness of the new dataset, new training strategies and the proposed model structure, as shown in Fig. 5. 'ESRGAN' represents the original ESRGAN method with GAN-based structure and DIV2KTimofte et al. (2017) (bicubic downsampling) as training data.

'ESRGAN*' represents the new-trained method with the same model structure but new datasets for training, including DIV2K Timofte et al. (2017) (Nearest neighbour downsampling) and GMSR. 'Ours' is the proposed method with the new model structure and new datasets for training. Comparison between 'ESRGAN' and 'ESRGAN*' can evaluate the effectiveness of new training datasets. While performance comparison between 'ESRGAN*' and 'Ours' evaluates the impacts in the model structure.

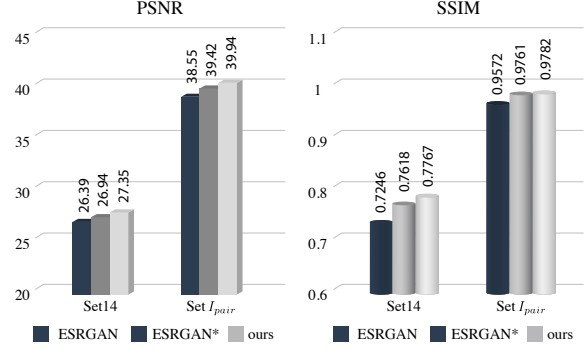

Figure 5: Comparing ESRGAN with the proposed method

PSNR and SSIM Wang et al. (2004) are utilized, where larger value represents better performance. The test was conducted on both general dataset set14 and the mobile image dataset Set $I_{pair}$. It can be seen that the 'ESRGAN*' outperforms 'ESRGAN' by changing training data. It proves that the new dataset(**GMSR**) and new training strategy can be easily generalized to other model structures and improve the performance on both general data and mobile data. Moreover, 'Ours' further improves the performance compared with 'ESRGAN*'. It proves the effectiveness of the model structure in $\Phi_{L2H}$.

## 5 CONCLUSION

In this work, a novel framework SRDGAN is proposed to solve the noise prior super resolution problem with dual generative adversarial network. A general degradation network H2L is proposed and is able to learn the noise prior of real-world LR image. By using our proposed training strategy, the H2L network is trained to generate the 'realistic' LR image paired with the HR image, and A large-scale benchmark general mobile super resolution dataset, **GMSR**, is generated from it. The **GMSR** can help super resolution methods to be well applied in the real-world mobile taken images. Meanwhile, a super resolution network L2H is proposed, which contains new structure and training strategies. Especially, these training strategies are proved to be effective for other SR model, such as ESRGAN, and improve the SR performance.

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
