# OpenReview forum: "SRDGAN: learning the noise prior for Super Resolution with Dual Generative Adversarial Networks"
_ICLR.cc/2020/Conference — Reject_

### Official Review · AnonReviewer1 · 2019-10-22
**Official Blind Review #1**

**Rating:** 3

**Review:**

In the paper, the authors proposed an end-to-end single image super-resolution framework, which is composed of two parts. First, a high-to-low network is trained to generate realistic HR/LR image pairs for training super-resolution models. Using this network, the authors designed a large-scale General Mobile Super Resolution Dataset, GMSR, which can be utilized for training or as a benchmark for super-resolution methods. Second, a low-to-high network is trained to produce a super-resolution image. Beneﬁting from the GMSR dataset and novel training strategies, the proposed low-to-high network could thus deal with the real distortions and restore ﬁne details.

The paper is easy to follow. The proposed framework is logical and the experiments proved to be effective. However, there are some problems as follows:

1. In the high-to-low network of framework, Gaussian random noise is added to clean high-resolution images to simulate the randomness of the distortions in LR. Why can do this in this way? The rationality of this setting should be given. Are there other kinds of noises be applied? As far as I knew, the in-camera processing pipeline will involve  Poisson-Gaussian noise rather than Gaussian only and other factors such as demosaicing, Gamma correction, and JPEG compression. It is hard to only consider Gaussian noise to stimulate the whole in-camera process.

2. The experimental part is remarkable insufficient to an ICLR paper. There only have a quantitative evaluation of the proposed framework with other super-resolution algorithms, while lacking the qualitative evaluation of the framework itself.

3. There are some formatting issues in the paper,
   1. Page 8, figure 5, here some errors in the label color of the figure.
   2. Page 1, introduction, line 14, "However, In most practical..." -> "However, in most practical...";
   3. Page 2, line 10, "...many problems, such as..., which is hard to avoid" -> "...many problems, such as..., which are hard to avoid";
   4. The last paragraph of the introduction, line 6, "...parameters.Third,..." -> "...parameters. Third,..."
      ...

**Experience Assessment:**

I have published one or two papers in this area.

**Review Assessment: Checking Correctness Of Derivations And Theory:**

N/A

**Review Assessment: Checking Correctness Of Experiments:**

I carefully checked the experiments.

**Review Assessment: Thoroughness In Paper Reading:**

I read the paper at least twice and used my best judgement in assessing the paper.

---

### Official Review · AnonReviewer2 · 2019-10-23
**Official Blind Review #2**

**Rating:** 3

**Review:**

This work proposes a framework for real world super resolution, which includes two networks: high-to-low and low-to-high. With high-to-low network, a general mobile super resolution dataset is proposed. According to the proposed framework and dataset, promising results are achieved for realistic image super-resolution.

[Strengths]
- This paper is well-written and easy to follow.
- A new dataset is proposed for real world image super-resolution task.

[Weaknesses]
- The novelty of this work is relatively limited. The idea to use a high-to-low network to model distribution of real-world low resolution images has already been investigated in several works, like Bulat et al. (2018) and Zhao et al. (2018). However, no comparisons are conducted with these two works. Besides, the network structures are very similar to SRGAN and ESRGAN, which again shows little novelty.

- The authors claimed the recent SOTA optical based method from Zhang et al. (2019) suffers from many problems. However, no direct comparisons are conducted to verify their viewpoints, and it's difficult to distinguish which one is better.

- Is there any visualization for the distributions of real work noisy images and the learned low resolution images? Besieds, since only one mobile device is used to capture the images, why does the proposed method generalize well on other mobile phones? Any distribution illustrations?

**Experience Assessment:**

I have published in this field for several years.

**Review Assessment: Checking Correctness Of Derivations And Theory:**

N/A

**Review Assessment: Checking Correctness Of Experiments:**

I carefully checked the experiments.

**Review Assessment: Thoroughness In Paper Reading:**

I read the paper thoroughly.

---

### Official Review · AnonReviewer3 · 2019-10-24
**Official Blind Review #3**

**Rating:** 1

**Review:**

This paper presents a single image super-resolution method. The motivation is real and practical.  In almost all existing methods the low-resolution training images are artificially synthesized using the available high-resolution images with bicubic downsampling, which allows LR images to carry more information than real demosaic-upscaled LR images. This mismatch between training and realistic
data hinders the practicability of such solutions (not limited to mobile scenarios).

Yet, the proposed remedy in the paper is not an answer either. It simply adds a noise image to the HR image, passes it through a convolutional network to construct an LR version, which is then fed into a deconvolutional network. Both these networks are build by residual blocks. Even worse, the paper simply uses zero-mean Gaussian random noise with a standard deviation of 0.05 to simulate the distortions. This is myopic and questionable.

The introduced dataset is very small, it contains only 447 pairs of images, with an additional 206 images as reference images. These pairs consist of a normal image (considered to be noisy) and a computed image by aggregating 20 images (considered to be clean) using a Blackberry Key2. In other words, the dataset only models a single device and a single camera lens. For practical purposes, this is not much different than using a bicubic sampling and adding noise drawn from a fixed distribution.

The paper fails to provide comparisons with the top methods in the NTIRE 2019 single image super-resolution challenge leaderboard.

**Experience Assessment:**

I have published in this field for several years.

**Review Assessment: Checking Correctness Of Derivations And Theory:**

I carefully checked the derivations and theory.

**Review Assessment: Checking Correctness Of Experiments:**

I carefully checked the experiments.

**Review Assessment: Thoroughness In Paper Reading:**

I read the paper thoroughly.

---

### Decision · Program_Chairs · 2019-12-19

**Decision:**

Reject

**Comment:**

All reviewers agree that the authors have done a great job identifying weaknesses with the current SOTA in super-resolution.   However, there is also agreement that the proposed approach may be too simple to accurately capture a range of real camera distortions, and more comparisons to the SOTA are needed.   While this paper certainly has merits and opens the door for strong work in the future, there is not enough support to accept the paper in its current form.